# Dysregulation of the DRAIC/SBK1 Axis Promotes Lung Cancer Progression

**DOI:** 10.3390/diagnostics14192227

**Published:** 2024-10-05

**Authors:** Rashed Alhammad, Milicia Allison, Fares Alhammad, Chinedu Anthony Anene

**Affiliations:** 1Department of Pharmacology, Faculty of Medicine, Kuwait University, Safat 13110, Kuwait; 2Centre for Cancer Biology and Therapy, School of Applied Science, London South Bank University, London SE1 0AA, UK; 3College of Science, Purdue University, West Lafayette, IN 47907, USA; 4Pediatrics Department, Sheikh Jaber Al-Ahmad Al-Sabah Hospital, Khalid Ben AbdulAziz Street, Sulaibikhat 13001, Kuwait; 5Centre for Cancer Genomics and Computational Biology, Barts Cancer Institute, Queen Mary University, London EC1M 6BQ, UK

**Keywords:** DRAIC, SBK1, bioinformatics, lung cancer, survival, lncRNA, miRNA

## Abstract

**Background:** Long non-coding RNAs (lncRNAs) are key regulators of cellular processes that underpin cancer development and progression. DRAIC is a migration inhibitor that has been linked with lung adenocarcinoma progression; however, its mechanisms remain to be studied. **Methods:** Several bioinformatics tools were used to explore the role of DRAIC in lung adenocarcinoma (LUAD) and lung squamous cell carcinoma (LUSC). **Results:** Our bioinformatics analysis illustrates that patients with low expression of DRAIC have poor overall survival outcomes. In addition, the mRNA of SH3 domain-binding kinase 1 (SBK1) was downregulated in this cohort of patients. Mechanistic analysis showed that SBK1 is under the DRAIC competing endogenous RNAs network, potentially through sponging of miRNA-92a. **Conclusions:** Consistent dysregulation of the DRAIC-SBK1 axis was linked to poor survival outcome in both LUAD and LUSC, suggesting a tumour inhibitor role and providing potential for new diagnostics and therapeutic approaches.

## 1. Introduction

Long non-coding RNAs (lncRNA) are greater than 200 nucleotides and include intergenic, intronic, sense, and antisense RNAs [1]. They have a wide range of roles in cell biology through regulating all aspect of gene expression [2]. Dysregulation of lncRNAs has been implicated in the development and progression of many human diseases, including cardiovascular disease, neurological disorder, and cancers [3,4,5]. The functionality of lncRNAs often depends on their cellular localisation, ranging from direct transcriptional regulation for nuclear localised transcripts to translational regulation for cytoplasmic transcripts [6]. Within the cytoplasm, lncRNAs have been shown to act as competing endogenous RNAs (ceRNAs). This regulatory mechanism involves lncRNAs competing with miRNAs for binding sites on target mRNAs. By binding to miRNA binding sites, lncRNAs can sequester miRNAs, preventing them from targeting and degrading their intended mRNA targets [2,7]. Here, several studies have described the mechanistic interactions that involve lncRNAs sequestering miRNAs to reduce their repression of target mRNAs [8,9,10]. The ability of lncRNAs to regulate gene expression allows cancer cells to subvert these networks towards survival and growth [11]. 

Lung cancer is one of the most common malignancies, with 2 million new cases and 1.8 million related global deaths yearly [12,13]. The mortality rate of lung cancer remains high due to lack of specific prognostic biomarkers to identify and treat patients at risk of disease progression. Recent studies have reported several lncRNAs that are altered in lung cancer [14,15,16]. The characterisation of prognosis-associated genes and their networks is required to effectively stratify and treat lung cancer patients. Several factors, such as KRAS, EGFR, and PDL1 have been identified as potential biomarkers for targeted therapeutics [17,18,19]. Dysregulated lncRNA networks could also serve as useful biomarkers for stratifying high-risk populations and the identification of tumours in the early stages of disease progression. Moreover, lncRNAs offer novel approaches for lung cancer treatment [20,21].

Herein, we identified a consistently altered ceRNA network in lung adenocarcinoma (LUAD) and lung squamous cell carcinoma (LUSC) that is associated with poor overall survival. DRAIC, which is a crucial part of this network, has been shown to play a role in lung cancer progression. It has been shown to mediate the proliferation, migration, and invasion in lung adenocarcinoma [22,23]. Moreover, DRAIC has been established for predicting prognosis in early-stage lung adenocarcinoma [24]. In contrast, other studies suggest that DRAIC might act as a tumour inhibitor [25,26]; however, the exact mechanisms remain to be elucidated. We demonstrate through bioinformatics analysis that the low expression of DRAIC is associated with a decrease in the levels of SBK1, potentially through the impaired sponging of miRNA-92a-1-5p. We further suggest that consistent alteration of this network has a negative impact on patient outcomes in LUSC and LUAD, highlighting its clinical significance.

## 2. Materials and Methods

### 2.1. Patient Cohorts

This study included 982 patients with lung cancer from two independent published cohorts: LUAD is the cancer genome atlas (TCGA) dataset that includes 507 treated lung adenocarcinoma patients, and LUSC is the cancer genome atlas (TCGA) dataset of 475 treated LUSC patients. We excluded cases with missing follow up, clinical variables and expression values. The expression of DRAIC across all TCGA tumours and normal tissue was investigated using the TIMER2.0 database (http://timer.cistrome.org/) (accessed on 25 August 2024) [27].

### 2.2. Survival Model and Differential Expression Analysis

Two strata lncRNA expression groups were defined by median stratification of the expression values (low < median, high > median), and hazard ratios (HRs) were calculated using univariable Cox PH models. The association of expression groups with altered expression of mRNA was assessed using the Student’s *t*-test, with *p* < 0.05 taken to indicate statistical significance. To investigate the biological processes associated with the altered mRNAs, the R ClusterProfiler package [28] and the R human annotation database (org.Hs.eg.db) were used to compare the ontologies of the differentially expressed genes [29]. Redundancies in enriched terms were reduced by strict q-value (<0.05) cut-off and semantic analysis.

### 2.3. Analysis of miRNA Targets

To identify validated mRNA targets of the DRAIC-associated miRNA, we extracted all their miRNA-mRNA interactions from the miRTarBase database [30]. Next, we extracted the mRNAs that were also differentially expressed in the DRAIC low expression group and used these for further analysis.

### 2.4. DRAIC Correlation with Tumour Stage, Lymph Nodes Stage, and Immune Infiltration

LUAD and LUSC datasets were downloaded from cBio Cancer Genomics Portal (http://cbioportal.org) (accessed 29 August 2024) (Appendix A) [31]. The distribution of expression level of DRAIC, tumour stage, and lymph nodes stage were measured across the samples. GEPIA was utilized to assess Pearson’s correlation coefficients between DRAIC and several gene markers of immune cells, including T cells, monocytes, macrophages, neutrophils, and dendritic cells [32]. Logrank *p*-values < 0.05 was considered statistically significant [33,34]. A *p*-value < 0.0001 was considered significant.

## 3. Results

### 3.1. Identification of Pan Lung Cancer Survival-Associated lncRNAs

LncRNAs are critical regulators of oncogenic programmes that underlie cancer hallmarks. Given the reported role in disease progression, we sought to identify pan-lung cancer prognostic lncRNAs in publicly available datasets of lung cancer samples. To this end, we stratified the expression (low and high: see Materials and Methods Section) of 542 lncRNAs measured in the TCGA LUSC dataset and analysed them by univariate Cox proportional hazards (PHs). We found eight lncRNAs associated with overall survival rates at *p* < 0.01 (Table 1). Specifically, the overexpression of LINC00667, DRAIC, LOC100130691, POLR2J4, and DSCR9 is favourable, while LINC01091, SPATA8, and FLJ36000 are unfavourable for LUSC patient survival (Table 1).

To determine if these lncRNAs have the same effect in LUAD, we extracted and stratified the expression of these eight lncRNAs in the TCGA LUAD dataset and repeated the univariate CoxPH (Table 2). DRAIC has a similar prognostic effect in LUSC and LUAD at *p* < 0.01 (Figure 1) but not the other seven lncRNAs (Table 1 and Table 2). Interestingly, we did not find mutations affecting the DRAIC gene in either dataset, suggesting DRAIC repression depends on transcriptional mechanisms.

Moreover, the differential expression of DRAIC between tumour and normal tissues across all TCGA tumours has been explored, with higher DRAIC expression being observed in breast invasive carcinoma (BRCA), LUAD, pheochromocytoma, paraganglioma (PCPG), prostate adenocarcinoma (PRAD), and thyroid carcinoma (THCA), compared to normal tissues (Figure 2). In contrast, lower DRAIC expression was observed in cholangiocarcinoma (CHOL), colon adenocarcinoma (COAD), glioblastoma multiforme (GMB), kidney chromophobe (KICH), kidney renal papillary cell carcinoma (KIRP), LUSC, and rectum adenocarcinoma (READ), compared to normal tissues (Figure 2).

### 3.2. DRAIC-Related mRNA Biological Processes

To inform the characterisation of the DRAIC regulatory network, we explored its sub-cellular localisation using lncLocator [35]. DRAIC was predicted to be localised in the cytoplasm (Score 0.42, Figure 3), suggesting that the function of DRAIC partially involves ceRNA mechanisms.

To determine whether altered DRAIC expression dysregulates specific mRNA, we first assessed which mRNAs were differentially expressed between DRAIC prognostic groups. We took this approach, as altered mRNA expression is a readout of deregulated ceRNA networks. DRAIC expression groups (low and high) were generated in both the LUSC and LUAD datasets, as described before (see Materials and Methods Section); we used the groups for differential expression analysis, using Student’s T-test for each dataset. The results highlighted 636 and 1503 differentially expressed mRNAs in LUSC and LUAD, respectively (Adjusted *p* value < 0.05). To investigate the biological processes associated with dysregulation of DRAIC in lung cancer, we applied gene ontology analysis to the sets of DE genes (LUSC and LUAD, independently). Ontology analysis revealed known cancer-associated processes for each set of DE genes, including extracellular matrix organisation, migration, cytokine signalling, immune response, and extracellular transport (Figure 4). Consistent with these observations, previous studies have shown that DRAIC acts as an inhibitor of cell migration. However, we observed only type-specific ontologies, suggesting that the integration of the two sets of DE genes will better capture the downstream DRAIC pathway, leading to its shared role in lung cancer progression. The integration of the two sets of genes (i.e., significant genes with the same direction in both LUAD and LUSC) highlighted 125 most consistently altered mRNAs, with the vast majority (107) downregulated in the low group (Appendix A). This observation suggests that the impaired sponging of miRNA by DRAIC may determine its role in lung cancer progression.

### 3.3. Identification of the Shared ceRNA Network

The competitive endogenous RNA (ceRNA) hypothesis proposes that transcripts with shared miRNA response elements compete for post-transcriptional control by miRNAs. Based on this mechanism of lncRNA activity, we set out to identify the level of ceRNA interactions within the 125 integrated genes. We identified 19 high confidence target miRNAs of DRAIC in humans using DIANA tools (https://diana.e-ce.uth.gr/lncbasev3/interactions) (accessed 20 February 2022), namely miR-100-5p, miR-135a, miR-135b-5p, miR-181b-5p, miR-2115-5p, miR-22-3p, miR-26a-5p, miR-30d-3p, miR-34a-5p, miR-363-3p, miR-365a-3p, miR-365b-3p, miR-509-3p, miR-660-5p, miR-671-5p, miR-9-5p, miR-92a-1-5p, and miR-99a-5p. Then, we utilised the miRTarBase database to predict where DRAIC can act as a sponge, sequestering miRNAs to regulate mRNA levels. We observed a high level of ceRNA interactions involving the DE genes in DRAIC high and low: 377 interactions with 18 miRNAs and 305 genes (94.7% predicted miRNA and 14% of DRAIC-associated DE genes). This observation is consistent with previous studies showing that DRAIC can indirectly regulate target protein-coding genes [36]. Focusing on the integrated DE genes, we found that six of the miRNAs co-interact with ten of the integrated genes (FOSL1, PLEKHB1, AKR7L, NT5E, PTBP3, ARNTL2, LRRC27, GFRA3, and SBK1), suggesting that these DE mRNAs and their ceRNA network are of biological significance.

### 3.4. DRAIC-miRNA-9a Dependent Repression of SBK1 Predicts Poor Survival

The ceRNA interactions of DRAIC suggest that low expression of this lncRNA may modulate these networks to promote lung cancer progression. Previous results have shown that mRNA targets of sponged miRNAs exhibit increased expression following the inhibition of the associated lncRNAs. Moreover, we found that the majority of the consistently altered mRNAs in the DRAIC expression group were downregulated, further suggesting that the deregulation of DRAIC ceRNA networks plays a role in the progression of lung cancer. To determine whether the DRAIC ceRNA networks specifically affected patient survival, we explored the impact of low expression of DRAIC combined with low expression of any of the network genes (FOSL1, PLEKHB1, AKR7L, NT5E, PTBP3, ARNTL2, LRRC27, GFRA3, and SBK1) on patient survival. In both LUAD and LUSC, we observed a significant (*p* < 0.01) poor survival outcome for patients with a low DRAIC and low SBK1 expression compared to the rest of the samples (Figure 5), implying that the DRAIC-SBK1 axis (predicted ceRNA interaction) consistently promotes lung cancer progression. These results were further supported by TargetScan analysis showing that miRNA-92a has nine binding sites in the SBK1 UTR, potentially degrading SBK1 mRNA through DRAIC dysregulation [37].

We further observed that low-DRAIC/low-AKR7L and low-DRAIC/low-GFRA3 were also associated with poor survival outcomes (*p* < 0.05) in both lung cancer types (Appendix A). Although the significance did not reach our strict threshold, the results suggests that the predicted miRNA (miR-34-5p) for both genes may support the role of DRAIC lung cancer progression. Moreover, five of the remaining genes showed significant results in at least one of the lung cancer types, including LRRC27 and PLEKHB1 in LUAD and ARNTL2, FOSL1, and PTBP3 in LUSC (Appendix A). These observations, combined with the type-specific enriched biological processes, suggests that low DRAIC expression could affect other ceRNA networks, leading to LUAD and LUSC progression.

### 3.5. DRAIC Correlates Negatively with Tumour Stage, Lymph Nodes Stage, and Immune Infiltration in LUAD and LUSC

The correlation between DRAIC and several clinical data was explored in LUAD and LUSC, in which significantly higher DRAIC expression was observed in tumour stage 1 compared to tumour stage 2 (Figure 6A) and in lymph node stage N0 compared to N1 (Figure 6B) in LUAD. In contrast, the difference in the expression of DRAIC in LUSC was not statistically significant (Figure 6C,D).

The correlation between the expression of DRAIC and immune infiltration was explored, as immune infiltration has been shown to correlate with the progression of LUAD and LUSC [38,39]. DRAIC is shown to negatively correlate with several markers of immune cells, including T cells, monocytes, macrophages, neutrophils, and dendritic cells in LUAD and LUSC (Figure 7).

## 4. Discussion

The functional analysis implicated lncRNAs in a wide range of cellular functions coordinated through transcriptional regulation, miRNA sponging, and translation regulation [1]. Their roles in cancer development and progression have been documented, including in lung cancer types [14]. Our analyses indicate that the dysregulation of the DRAIC-SBK1 axis might also play a role in lung cancer progression. We demonstrate that low expression of DRAIC is consistently associated with poor overall survival outcomes for lung cancer patients, highlighting the importance of lncRNA-signalling networks. Furthermore, we explored the biological significance of the ceRNA network associated with DRAIC in lung cancer samples and identified SBK1 as an important component of this prognostic utility, potentially through miRNA-92a-1-5p (Figure 4). Previous studies have indicated that DRAIC is capable of miRNA sponging, including the miRNA-92a family, miR-34a, and other oncogenic miRNAs [25,40]. Many of these miRNAs are characterised as tumour promoters, implicating DRAIC as suppressive tumour lncRNA. Consistently, DRAIC has been shown to inhibit migration and proliferation, albeit in prostate cancer cells [41]. Additionally, DRAIC is found to be downregulated in several cancers, including gastric, lung, and liver [36,42]. Oncogenic mutations affecting certain transcription factors have been shown to dysregulate the expression of lncRNAs, including DRAIC [40], which affects downstream ceRNA networks [43].

LncRNA functionality is linked to their localisation, with cytoplasm-transported lncRNAs interacting with protein-coding mRNAs or acting as translational regulators [6,44]. Using a sequence-based prediction tool, DRAIC was shown to be localised in the cytoplasm, indicative of a ceRNA-based function. Several published reports agree with our analysis, in which DRAIC was shown to be mostly localized in the cytoplasm of cancer cells [41,45,46]. This was further supported by the identification of several ceRNA interactions in both the LUAD- and LUSC-altered mRNAs. Of particular significance is that matched low expression of DRAIC and SBK1 mRNAs was associated with poor survival in both lung cancer types. Indeed, SBK1 has been implicated in multiple cancers but exhibits cancer-specific expression and effects [47]. It is upregulated in ovarian cancers [47], while having the inverse expression in the oesophagus and in lung cancer [47]. Previous studies on ovarian cancer cell lines reported a survival-promoting effect through protection of cancer cells from apoptosis [47]. However, in cervical cancer, its high expression is favourable for survival [48]. Consistently, we found that the DRAIC-associated downregulation of SBK1 is unfavourable for lung cancer survival.

Given that our previous results suggest that DRAIC might act as a tumour inhibitor in lung cancer, the correlations between DRAIC, clinical data, and immune infiltration were investigated to further validate our observations. The negative correlation between DRAIC and several markers of immune cells in LUAD and LUSC suggests that DRAIC might play a role as a tumour inhibitor, as immune infiltration correlates with the progression and development of LUAD [38]. Moreover, the negative correlation between DRAIC expression and tumour stage in LUAD supports our hypothesis, that DRAIC might play a tumour-inhibitor role in LUAD, as higher DRAIC expression was observed in early-stage compared to late-stage LUAD. Moreover, our results showed that DRAIC negatively correlates with the lymph node stage, suggesting that it might act as an inhibitor of cell migration. This agrees with our ontology analysis and with several published reports, suggesting that DRAIC might act as an inhibitor for cell migration [25,26].

Despite the above results and the advantages of such bioinformatics analysis, some limitations in the results need to be addressed. Specifically, the mechanism by which the loss of SBK1 promotes lung cancer progression remains unknown. Further mechanistic analysis of SBK1 using in vitro and animal models is needed to robustly elucidate these mechanisms. Such analysis will contribute to the development of novel therapeutics specifically targeted at patients with downregulated DRAIC.

## 5. Conclusions

In summary, we show for the first time the pan-lung cancer prognostic role of DRAIC downregulation, potentially through dysregulated miR-92a-1-5p and the corresponding downregulation of SBK1. We identified other mRNA targets of DRAIC, which may contribute to lung cancer progression. The finding of this bioinformatics analysis suggests that targeting DRAIC ceRNA may hold both prognostic and therapeutic utility in lung cancer management.

## Figures and Tables

**Figure 1 diagnostics-14-02227-f001:**
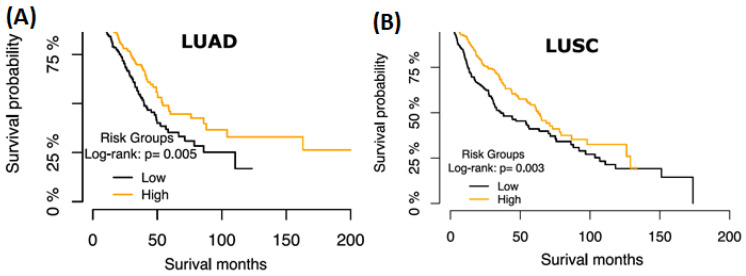
DRAIC regulatory network in lung cancer progression. (**A**) Kaplan–Meier plot of survival probabilities over time for patients stratified on the DRAIC expression groups (high vs. low) in LUAD. (**B**) Kaplan–Meier plot of survival probabilities over time for patients stratified on the DRAIC expression groups (high vs. low) in LUSC.

**Figure 2 diagnostics-14-02227-f002:**
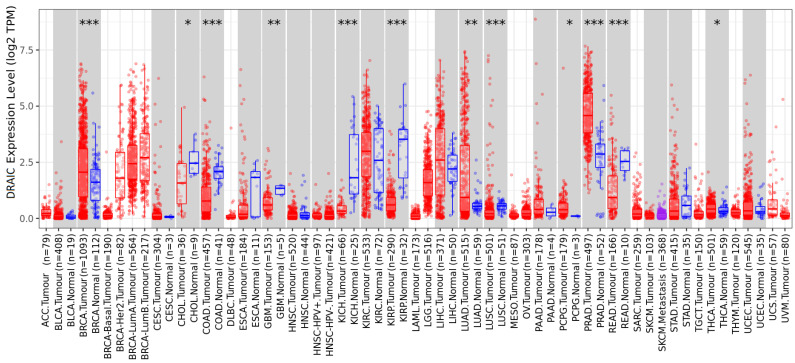
This shows differential expression of DRAIC between tumour and adjacent normal tissues across all TCGA tumours. Distributions of DRAIC gene expression levels are displayed using box plots. The statistical significance computed by the Wilcoxon test is annotated by the number of stars (*: *p*-value < 0.05; **: *p*-value < 0.01; ***: *p*-value < 0.001).

**Figure 3 diagnostics-14-02227-f003:**
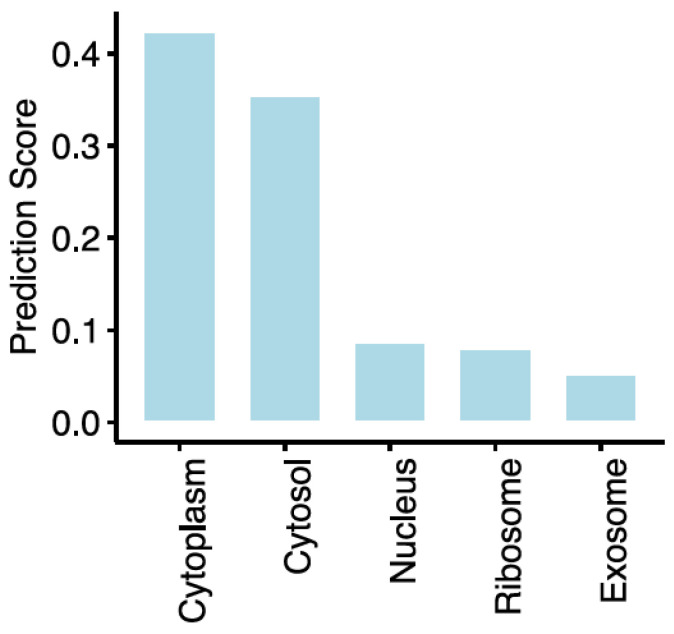
DRAIC regulatory network in lung cancer progression. Bar plot of DRAIC expression score across subcellular locations.

**Figure 4 diagnostics-14-02227-f004:**
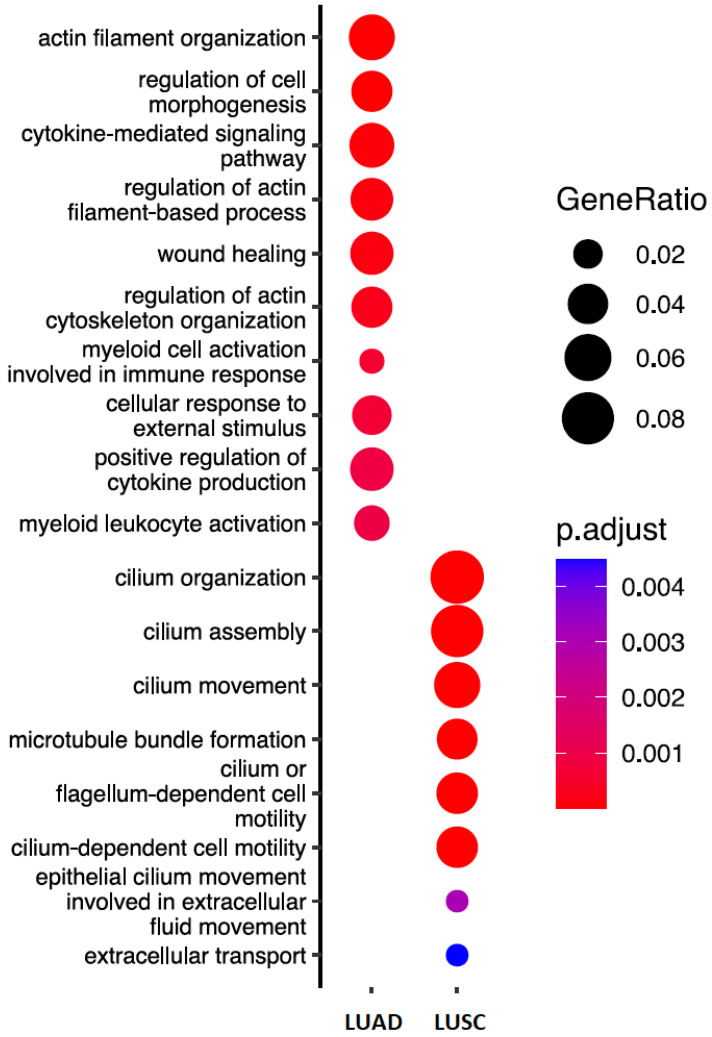
DRAIC regulatory network in lung cancer progression. Gene ontology analysis of biological processes associated with DE genes in DRAIC expression groups for both lung cancer types.

**Figure 5 diagnostics-14-02227-f005:**
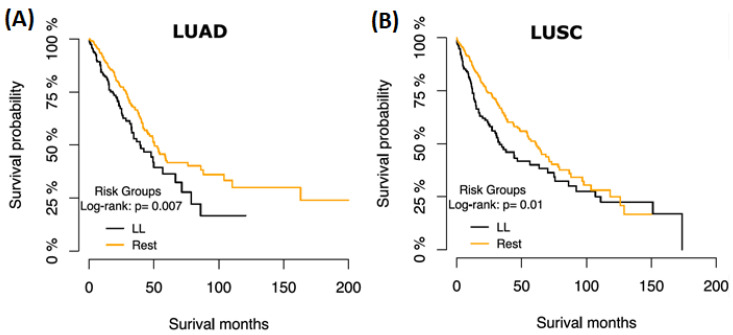
DRAIC-SBK1 downregulation predicts poor lung cancer patient survival. (**A**) Kaplan–Meier plot of survival probability over time for patients stratified by the combined DRAIC and SBK1 expression groups in LUAD. (**B**) Kaplan–Meier plot of survival probability over time for patients stratified on the combined DRAIC and SBK1 expression groups in LUSC.

**Figure 6 diagnostics-14-02227-f006:**
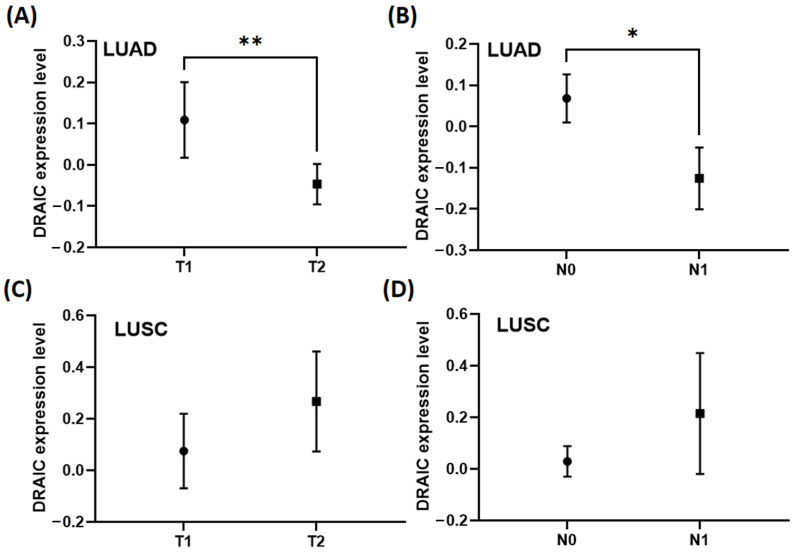
DRAIC expression in tumour stages and lymph nodes stages. (**A**) DRAIC expression in tumour stage 1 vs. tumour stage 2 in LUAD. (**B**) DRAIC expression in lymph node stage N0 (no cancer in nearby lymph nodes) vs. N1 (cancer is found in lymph nodes) in LUAD. (**C**) DRAIC expression in tumour stage 1 vs. tumour stage 2 in LUSC. (**D**) DRAIC expression in lymph node stage N0 vs. N1 in LUSC. * *p* < 0.05 and ** *p* < 0.01.

**Figure 7 diagnostics-14-02227-f007:**
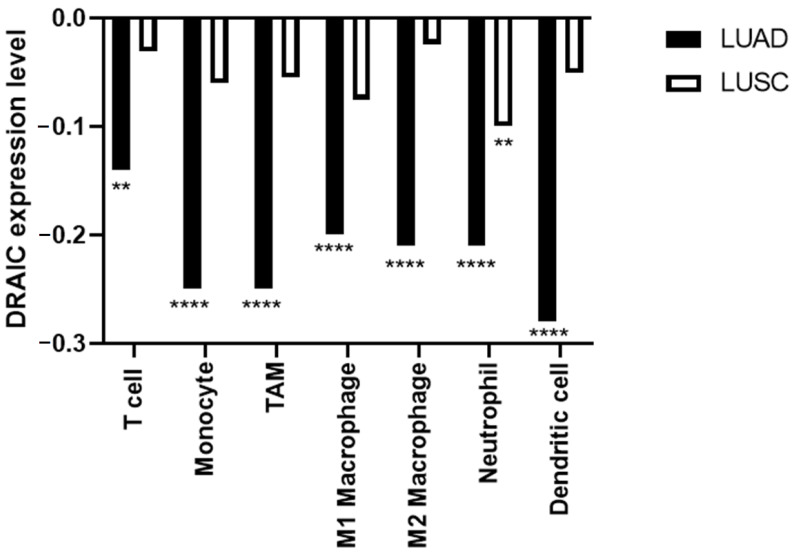
Pearson’s correlation coefficient between DRAIC and several gene markers of immune cells in LUAD and LUSC. ** *p* < 0.01 and **** *p* < 0.0001.

**Table 1 diagnostics-14-02227-t001:** List of prognostic lncRNA RNAs in LUSC, showing survival analysis results.

LncRNA	Beta	HR * (95% CI **)	Log.Test	*p*-Value
POLR2J4	−0.38	0.68 (0.52–0.9)	7.4	0.00641
DRAIC	−0.41	0.66 (0.5–0.87)	8.9	0.0028
SPATA8	0.37	1.5 (1.1–1.9)	7.3	0.00691
DSCR9	−0.35	0.7 (0.54–0.93)	6.3	0.0118
FLJ36000	0.36	1.4 (1.1–1.9)	6.4	0.0115
LINC01091	0.39	1.5 (1.1–1.9)	7.9	0.00494
LINC00667	−0.43	0.65 (0.49–0.85)	9.7	0.00189
LOC100130691	−0.38	0.68 (0.52–0.89)	7.7	0.00564

* Hazard ratio. ** Confidence interval.

**Table 2 diagnostics-14-02227-t002:** List of prognostic lncRNA RNAs in LUAD, showing survival analysis results.

LncRNA	Beta	HR * (95% CI **)	Log.Test	*p*-Value
LINC00667	−0.17	0.84 (0.63–1.1)	1.4	0.239
DRAIC	−0.41	0.66 (0.49–0.89)	7.7	0.00553
POLR2J4	0.042	1 (0.78–1.4)	0.078	0.78
DSCR9	0.042	1 (0.78–1.4)	0.081	0.776
LINC01091	−0.3	0.74 (0.55–0.99)	4.2	0.0399
SPATA8	0.066	1.1 (0.8–1.4)	0.19	0.659

* Hazard ratio. ** Confidence interval.

## Data Availability

The datasets generated during the current study are available from the corresponding author on reasonable request.

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
