# Peer review of "Dysregulation of the DRAIC/SBK1 Axis Promotes Lung Cancer Progression"

_diagnostics, 2024, doi:10.3390/diagnostics14192227_

Round 1

Reviewer 1 Report (Previous Reviewer 4)

Comments and Suggestions for Authors

In this manuscript, the authors investigate the DRAIC in lung cancer, conduct the survival analysis. They identify SBK1 is under the DRAIC competing 19 endogenous RNAs network, potentially through sponging of miRNA-92a. The authors have improved the quality of the results and revised their manuscript according my previous comments. Their current works may be acceptable to publish in our journal, but the following revisions should be completed.

Major comments
1. In lines 71-73, it is described that “within the cytoplasm, lncRNAs have been shown to act as competing endogenous RNAs (ceRNAs)”. It is suggested to add an introduction to the regulation mechanism of ceRNAs before this sentence. In addition, the following sentence “when lncRNAs compete with other non coding RNAs for mRNA binding and regulation” need to re-write.

2. Since Tables 1 and 2 are studies on lncRNAs, using genes in the header is not appropriate and can easily lead to misunderstandings and confusion for readers.

3. Compare Table 1 and Table 2 to see if it would be better to place them after Table 1. Then adjust the position of Figure 2 and Table 2 synchronously.

4. Table 1 indicates that overexpression of five lncRNAs, LINC00667, DRAIC, LOC100130691, POLR2J4, and DSCR9, is beneficial. But why was only the differential expression of DRAIC analyzed between different tumors in Figure 1, or why was DRAIC selected for differential expression analysis between different tumors.

5. Please explain why after analyzing 8 lncRNAs, the results presented in Table 2 only show 6 lncRNAs.

6. A higher DRAIC expression observed in LUAD and a lower DRAIC expression observed in LUSC, but why the similar prognostic effects (survival time) were observed in Figure 2 for LUSC and LUAD?

7. Since Figure 2 is intended to compare whether DRAIC has similar prognostic effects (survival time) for LUSC and LUAD, the position distribution of the four subgraphs in Figure 2 is not appropriate, which is not conducive to readers observing whether LUSC and LUAD are similar. Therefore, their position distribution should be adjusted by placing subgraphs with the same p-value together instead of randomly placing them.

Author Response

  1. In lines 71-73, it is described that “within the cytoplasm, lncRNAs have been shown to act as competing endogenous RNAs (ceRNAs)”. It is suggested to add an introduction to the regulation mechanism of ceRNAs before this sentence. In addition, the following sentence “when lncRNAs compete with other non coding RNAs for mRNA binding and regulation” need to re-write.

The following sentences were added: “Within the cytoplasm, lncRNAs has been shown to act as competing endogenous RNAs (ceRNAs). This regulatory mechanism involves lncRNAs competing with miRNAs for binding sites on target mRNAs. By binding to miRNA binding sites, lncRNAs can sequester miRNAs, preventing them from targeting and degrading their intended mRNA targets [2], [7].”

  1. Since Tables 1 and 2 are studies on lncRNAs, using genes in the header is not appropriate and can easily lead to misunderstandings and confusion for readers.

The headers were changed.

  1. Compare Table 1 and Table 2 to see if it would be better to place them after Table 1. Then adjust the position of Figure 2 and Table 2 synchronously.

The positions of the tables and Figures were adjusted.

  1. Table 1 indicates that overexpression of five lncRNAs, LINC00667, DRAIC, LOC100130691, POLR2J4, and DSCR9, is beneficial. But why was only the differential expression of DRAIC analyzed between different tumors in Figure 1, or why was DRAIC selected for differential expression analysis between different tumors.

Although Table 1 indicates that overexpression of five IncRNAs is beneficial in LUSC, only DRAIC showed the same pattern in LUAD in Table 2. Hence, DRAIC was selected for further analysis. Moreover, DRAIC was analysed across all TCGA tumour as one of the reviewers asked to do this analysis.

  1. Please explain why after analyzing 8 lncRNAs, the results presented in Table 2 only show 6 lncRNAs.

Our initial analysis was carried out on 542 lncRNAs, in which univariate Cox proportional hazards analysis was performed in LUSC. We found eight lncRNAs associated with overall survival rates at p < 0.01 (Table 1). Then, we extracted and stratified the expression of these eight lncRNAs in LUAD dataset and repeated the univariate CoxPH. The data for LOC100130691 and FLJ36000 in LUAD dataset was not available, hence only the 6 IncRNAs were included in Table 2.

  1. A higher DRAIC expression observed in LUAD and a lower DRAIC expression observed in LUSC, but why the similar prognostic effects (survival time) were observed in Figure 2 for LUSC and LUAD?

Although different DRAIC expression was observed in LUAD and LUSC compared to normal tissue, DRAIC showed similar prognostic effect in both LUAD and LUSC. This could be due to various reasons, such as DRAIC could be involved in regulatory networks that influence survival pathways in both cancers, regardless of its expression level. Moreover, other signalling pathways or genes may compensate for DRAIC’s expression changes, leading to similar prognostic outcomes. This could involve other lncRNAs, miRNAs, or proteins that interact with DRAIC.

  1. Since Figure 2 is intended to compare whether DRAIC has similar prognostic effects (survival time) for LUSC and LUAD, the position distribution of the four subgraphs in Figure 2 is not appropriate, which is not conducive to readers observing whether LUSC and LUAD are similar. Therefore, their position distribution should be adjusted by placing subgraphs with the same p-value together instead of randomly placing them.

The distribution of the subgraphs were adjusted to make it easier for the reader.

Reviewer 2 Report (Previous Reviewer 2)

Comments and Suggestions for Authors

OK

Author Response

N/A

Round 2

Reviewer 1 Report (Previous Reviewer 4)

Comments and Suggestions for Authors

The authors have improved their manuscript according to my comments, I suggest accept in present form.

This manuscript is a resubmission of an earlier submission. The following is a list of the peer review reports and author responses from that submission.

Round 1

Reviewer 1 Report

Comments and Suggestions for Authors

The manuscript entitled "Dysregulation of the DRAIC/SBK1 axis promotes lung cancer progression" reports bioinformatics findings from analyzing public databases for DRAIC and related molecules. Although it lacks wet-lab results, the findings may provide useful information for the field to use DRAIC and lncRNAs in lung cancer prognosis.

The writing of the manuscript is acceptable and does not need major language revisions. However, the authors are strongly recommended to read it several times more to make sure no errors will appear eventually.

Before this manuscript can be recommended by this reviewer for acceptance, the authors will need to address the following points:

1. Many statements are too general and lack specifics. For example, the 1st sentence in the Abstract, "Identification of novel molecules and pathways affecting lung cancer 12 progression is an unmet clinical need."; what did the authors want to express exactly? Many novel molecules have been actually reported, so how could that become an unmet clinical need?! Another example is in the discussion section: "Here, we show that dysregulation of the DRAIC-SBK1 206 axis also plays a role in lung cancer progression." How did the authors make this conclusion? They did not provide any functional data, so this claim is not actually merited with just statistical analysis results. Therefore, the authors must modify their writing to make sure all claims are backed by real data in the manuscript.

2. Perform at least one validation experiment, e.g., do a quick staining to show the sub-cellular localization patterns claimed by Figure 2. Or the authors could perform some functional studies to show DRAIC or related molecules do have roles in lung cancer progression.

Author Response

The authors present an interesting study that warrants a number of explanations given the reviewers' comments. In order to reconsider this manuscript, the authors must address all the key issues raised by the reviewers, as noted below:

  1. Many statements are too general and lack specifics. For example, the 1st sentence in the Abstract, "Identification of novel molecules and pathways affecting lung cancer 12 progression is an unmet clinical need."; what did the authors want to express exactly? Many novel molecules have been actually reported, so how could that become an unmet clinical need?! Another example is in the discussion section: "Here, we show that dysregulation of the DRAIC-SBK1 206 axis also plays a role in lung cancer progression." How did the authors make this conclusion? They did not provide any functional data, so this claim is not actually merited with just statistical analysis results. Therefore, the authors must modify their writing to make sure all claims are backed by real data in the manuscript.

The writing was modified to make sure all claims are backed by real data in the manuscript.

  1. Perform at least one validation experiment, e.g., (a) do a quick staining to show the sub-cellular localization patterns claimed by Figure 2, (b) Or the authors could perform some functional studies to show DRAIC or related molecules do have roles in lung cancer progression.
  • The sub-cellular localization patterns claimed by Figure 2 has been shown in several published reports indicating that DRAIC is mostly localized in the cytoplasm of cancer cells [1], [2], [3].

  • Several published reports showed that DRAIC plays a role in lung cancer progression, in which it has been shown to mediate the proliferation, migration, and invasion in lung adenocarcinoma [4], [5]. Moreover, DRAIC has been established for predicting prognosis in early-stage lung adenocarcinoma [6]. In addition, the correlations between DRAIC and several clinical data were added to the manuscript (figures 6A-6D and 7).

Reviewer 2 Report

Comments and Suggestions for Authors

NICE WORK!VERY INTERESTING.FEW THINGS TO TAKE UNDER CONSIDERATION: 1)AT ABSTRACT(PG1)EXPLAIN LUAD,LUSC,2)INTRODUCTION LINE 46:EGFR,PD-L1 ARE ALLREADY TARGETABLE,KRAS IS POTENTIAL TARGETABLE.

IN GENERAL ,SEE IF THERE IS A CORRELATION BETWEEN LOW DRAIC AND SUBTYPE OR GRADE OF THE CANCER

Author Response

See if there is a correlation between low DRAIC and subtype or grade of the cancer.

The correlation between DRAIC, tumour stage, lymph nodes, and immune infiltration were added to the manuscript in Figures 6(A-D) and 7.

Reviewer 3 Report

Comments and Suggestions for Authors

I think the researcher's experiment design is reasonable and the methods are correct. This study is suitable for clinical practice and has great significance. I suggest publication. 

Reviewer 4 Report

Comments and Suggestions for Authors

In this manuscript, the authors investigate the DRAIC in lung cancer, conduct the survival analysis. They identify SBK1 is under the DRAIC competing 19 endogenous RNAs network, potentially through sponging of miRNA-92a.

However, in my opinion, the analysis of one lncRNA in one type of cancer is too simple. The works are not enough for publishing in the Diagnostics journal (an IF=3.992 journal). Compared with similar works published in other SCI journals, the authors at least need to complete one of the following works:

(1) analyze DRAIC in all available types of cancers in TCGA

(2) analyze at least 10 key genes (including DRAIC) in lung cancer

(3) analyze DRAIC in lung cancer and validated by wet lab

 Another comments for the authors:

1.     Because the authors analyze DRAIC, however, both in the Abstract and Introduction parts, the authors not describe why choose DRAIC and how DRAIC is important in lung cancer?

2.     The citation “[12], [13]” maybe need to change to “[12-13]”; “[14]–[16]” change to “[14-16]”

3.  Table 1 Legend, “LUCS”? maybe change to “LUSC”

4.  Line 102,  “LSCC”, mistyped again

5.  Redundant description: Line 99 “lung adenocarcinoma (LUAD)”, again in Line 101 “lung adenocarcinoma (LUAD)”

Author Response

The analysis of one lncRNA in one type of cancer is too simple. The authors need to complete one of the following experiments: (1) analyze DRAIC in all available types of cancers in TCGA (2) analyze at least 10 key genes (including DRAIC) in lung cancer (3) analyze DRAIC in lung cancer and validated by wet lab.

  • Our initial analysis was carried out on 542 lncRNAs, in which univariate Cox proportional hazards analysis was performed in LUSC. We found eight lncRNAs associated with overall survival rates at p < 0.01 (Table 1). Then, we extracted and stratified the expression of these eight lncRNAs in LUAD dataset and repeated the univariate CoxPH. Only DRAIC showed similar in LUSC and LUAD.

  • In addition, we have analysed DRAIC in all available types of cancers in TCGA, in which Figure 1 was added to the manuscript showing the differential expression of DRAIC between tumour and adjacent normal tissues across all TCGA tumours

Other minor comments for the authors: 1. Because the authors analyze DRAIC, however, both in the Abstract and Introduction parts, the authors do not describe why the chose DRAIC and how important is DRAIC in lung cancer? 2. The citation “[12], [13]” maybe need to change to “[12-13]”; “[14]–[16]” change to “[14-16]” 3. Table 1 Legend, “LUCS”? maybe change to “LUSC” 4. Line 102, “LSCC”, mistyped again 5. Redundant description: Line 99 “lung adenocarcinoma (LUAD)”, again in Line 101 “lung adenocarcinoma (LUAD)”

  1. We have added why we chose DRAIC and how important is DRAIC in lung cancer.

  1. The citations were changed

  1. Table legend was changed

  1. The typo errors were corrected

References

[1]        K. Sakurai, B. J. Reon, J. Anaya, and A. Dutta, ‘The lncRNA DRAIC/PCAT29 locus constitutes a tumor suppressive nexus’, Mol Cancer Res, vol. 13, no. 5, pp. 828–838, May 2015, doi: 10.1158/1541-7786.MCR-15-0016-T.

[2]        ‘Chromogenic in situ hybridization reveals specific expression pattern of long non-coding RNA DRAIC in formalin-fixed paraffin-embedded specimen - PubMed’. Accessed: Aug. 19, 2024. [Online]. Available: https://pubmed.ncbi.nlm.nih.gov/38075206/

[3]        M. Sun, S. S. Gadad, D.-S. Kim, and W. L. Kraus, ‘Discovery, Annotation, and Functional Analysis of Long Noncoding RNAs Controlling Cell-Cycle Gene Expression and Proliferation in Breast Cancer Cells’, Molecular Cell, vol. 59, no. 4, pp. 698–711, Aug. 2015, doi: 10.1016/j.molcel.2015.06.023.

[4]        ‘Prognostic Value of lncRNA DRAIC and miR-3940-3p in Lung Adenocarcinoma and Their Effect on Lung Adenocarcinoma Cell Progression - PMC’. Accessed: Aug. 26, 2024. [Online]. Available: https://www.ncbi.nlm.nih.gov/pmc/articles/PMC8577463/

[5]        Liu B.-L. et al., ‘LncRNA DRAIC regulates the proliferation, apoptosis, migration and invasion of lung adenocarcinoma cells by targeting let-7i-5p’, Chinese Journal of Oncology, pp. 471–481, 2023.

[6]        L. Mu, K. Ding, R. Tu, and W. Yang, ‘Identification of 4 immune cells and a 5-lncRNA risk signature with prognosis for early-stage lung adenocarcinoma’, J Transl Med, vol. 19, p. 127, Mar. 2021, doi: 10.1186/s12967-021-02800-x.
